# Control Over the Morphology of Electrospun Microfibrous Mats of a Polymer of Intrinsic Microporosity

**DOI:** 10.3390/membranes11060422

**Published:** 2021-05-31

**Authors:** Elsa Lasseuguette, Richard Malpass-Evans, John M. Tobin, Neil B. McKeown, Maria-Chiara Ferrari

**Affiliations:** 1School of Engineering, University of Edinburgh, Robert Stevenson Road, Edinburgh EH9 3FB, UK; M.Ferrari@ed.ac.uk; 2EaStCHEM School of Chemistry, University of Edinburgh, David Brewster Road, Edinburgh EH9 3FJ, UK; R.Malpass-Evans@ed.ac.uk (R.M.-E.); J.Tobin@ed.ac.uk (J.M.T.); Neil.McKeown@ed.ac.uk (N.B.M.)

**Keywords:** electrospinning, microfibres, PIMEATB, porosity, lactate, breathability, waterproof

## Abstract

This study reports for the first time the preparation of an electrospun microfibrous mat of PIM-EA-TB. The electrospinning was carried out using a chloroform/n-Propyl-lactate (n-PL) binary solvent system with different chloroform/nPL ratios, in order to control the morphology of the microfibres. With pure chloroform, porous and dumbbell shape fibres were obtained whereas, with the addition on n-PL, circular and thinner fibres have been produced due to the higher boiling point and the higher conductivity of n-PL. The electrospinning process conditions were investigated to evaluate their impact on the fibres’ morphology. These microfibrous mats presented potential to be used as breathable/waterproof materials, with a pore diameter of 11 μm, an air resistance of 25.10^−7^ m^−1^ and water breakthrough pressure of 50 mBar.

## 1. Introduction

Electrospinning is a straightforward method to produce self-standing microfibre membranes presenting high porosity and pore size ranging from ten nanometres to several micrometres [1]. Microfibres are defined as a continuous filament with an average fibre diameter of 25 microns or smaller and have been mainly used for wastewater treatment [2,3], smart responsive surface [4,5], and bioengineering application [6]. They present larger pores which can allow or facilitate cellular infiltration and/or diffusion of nutrients in vitro culture [7]. The generation of fibres is based on the formation of a jet from a charged polymeric system under an electrical field. The solvent evaporation and the stretching of the jet, caused by the repulsive forces of the charged molecules within the jet, are responsible for the formation of the polymer fibres [8,9,10,11,12]. Thus, the final mat fibrous morphology depends on the polymer–solvent system (such as solvent nature, viscosity, conductivity) and on the electrospinning process parameters (such as feed flow rate, voltage, distance between the tip and the collector). Therefore, various nanostructures, from beads to bead-free fibres with different fibre diameter can be produced by tuning these parameters.

Polymers of Intrinsic Microporosity (PIMs) are a class of macromolecule which have generated considerable interest in the field of gas separation [13,14,15,16,17,18,19,20], hydrogen storage [21,22], sensors [23] or liquid separation [22,24] thanks to a high surface area (typically 300–1500 m^2^/g) and interconnected micropores (<2 nm) [15]. Another advantage of PIMs is their solution processability in common organic solvents which allow the optimisation of the macroscopic formation of the microporous polymer, for example as electrospun fibres, coatings on woven fabrics, etc. Among others, research on electrospun PIMs fibres, mainly on PIM-1 and on modified PIM-1, has been gaining momentum in recent years [6,12,25,26,27,28]. Recently, a new type of PIM, derived from ethanoanthracene and Tröger’s base (PIM-EA-TB), has been developed using a polymerisation reaction based on the formation of the bridged bicyclic diamine called Tröger’s base (TB: 6H,12H-5,11-methanodibenzo[b,f][1,5]diazocine) [14,29]. PIM-EA-TB contains only benzene rings fused together via rigid bridged bicyclic units composed of TB and ethanoanthracene EA and demonstrates an apparent higher BET surface area than PIM-1 [14]. Its high rigidity results in enhanced gas separation performance with a superior vapour sorption capability [15]. Moreover, thanks to its basicity with the Tröger’s base, PIM-EA-TB can act as an ion-conducting material and can be used to construct ionic diodes [30]. To the best of our knowledge, no research on the electrospinning of PIM-EA-TB has been published.

Here, we report for the first time the fabrication of PIM-EA-TB fibres with controlled morphology by an electrospinning process using a binary solvent system. The aim of this study is to investigate the impact of the different solvent systems properties and the electrospinning process conditions (feed flow rate, voltage, distance between tip and collector) on the surface morphology and diameter of electrospun PIM-EA-TB fibres. The resulting fibres were further characterised to assess their potential for application in terms of waterproof resistance, breathability and air permeability.

## 2. Materials and Methods

### 2.1. Materials

PIM-EA-TB (Figure 1a) was prepared and characterised using a previously reported procedure [25].

Chloroform was purchased from VWR and used without purification. n-Propyl lactate was kindly provided by Corbion and used without purification. The physical properties of each solvent are shown in Table 1.

### 2.2. Electrospinning Technology

Electrospinning of PIM-EA-TB was achieved by using the apparatus (IME Technologies, Waalre, Netherlands) with 20% wt/vol solutions created using a binary solvent system with different chloroform/n-PL ratios and stirred for approximately 4 h prior to processing to ensure thorough mixing. Electrospun mats (Figure 1b) were generated as described previously [26] in the horizontal electrospinning setup shown, with various flow rates, operating distances, and applied voltages.

The physical properties of each solutions are presented in Appendix A.

### 2.3. Characterisation Methods

#### 2.3.1. Scanning Electron Microscopy (SEM)

The fibrous mats were analysed with a JSM-IT100 (JEOL, Tokyo, Japan) operating at 10 kV after being treated by sputtering a 9 nm layer of gold to form a conductive surface. The mean diameter (OD) of electrospun fibres was determined using the ImageJ software. The mean fibre diameter was determined from a minimum of 30 measurements of the random fibres in three SEM images taken from different areas of the mat.

#### 2.3.2. Measurement of Pore Size Diameter (r), Hydrostatic Pressure (P_H2O_) and Air Flow Resistance (R_air_)

The pore size diameters (r), hydrostatic pressure (P_H2O_) and air flow resistance (R_air_) of the membranes were measured using Quantachrome Porometer3Gzh (Anton Paar GmbH, St. Albans, UK). For the pore size diameters (r) and the hydrostatic pressure (P_H2O_), the sample was initially wetted by a liquid with low surface tension (Porometer 3G—Porofil^®^ Wetting Solution) and water, respectively, and an increasing pressure of air was applied (from 0.01 to 1.4 bar). P_H2O_ corresponds to the pressure when the initial flow (i.e., the bubble point; at which gas is first seen to pass through the specimen) is detected by the apparatus.

The airflow resistance (R_A__ir_) was measured with the dry flow method. In this mode, the instrument measures the flow of air through a dry porous medium for a known pressure drop. The air permeability, k_air_, is calculated from the Equation (1):(1)$$k_{\mathrm{air}}=\frac{\eta \times Q}{A}\times \frac{\Delta x}{\Delta P}$$
where η is the dynamic viscosity of air (1.83 × 10^−5^ kg·m^−1^ s^−1^), Q is the airflow measured through the sample, A is the sample area, Δx is the sample thickness, and ΔP is the pressure drop. However, on fibrous mats, thickness measurement is often problematic and can be a large source of error. It is preferred to present the pressure drop/flow rate results in terms of an apparent flow resistance defined by Equation (2) [31].
(2)$$R_{\mathrm{Air}}=\frac{A\times \Delta P}{\eta \times Q}$$
where $R_{\mathrm{Air}}$ is the apparent Darcy airflow resistance in m^−1^.

#### 2.3.3. Conductivity Measurement

The conductivity of the samples was measured by means of a SciQuip conductivity meter (SciQuip Inc., Manchester, UK).

## 3. Results

### 3.1. Effect of the Process Conditions

The aim of this study is to fabricate fibrous PIM-EA-TB with a regular diameter and without defects, i.e., with no bead-like structure. As, the morphology and porosity of the fibrous mat can be tuned by adjusting the process conditions, we investigate the influence of solvent, flow rate, distance between the tip and the collector and voltage.

#### 3.1.1. Effect of Solvent

The electrospinning of PIM-EA-TB microfibers was carried out using a chloroform/n-PL binary solvent system with different chloroform/n-PL ratios, adjusted from 10:0, 9:1, 7:3, 5:5, 3:7 to 0:10 (*v*/*v*). The concentration of PIM-EA-TB was kept constant at 20% wt/vol. At lower concentration, spraying was noticed whereas at higher concentration, the solution was too viscous and induced a blockage of the syringe.

Morphology

The SEM images of the PIM-EA-TB fibres in different solvent systems are shown in Figure 2. It was observed that the morphology of the electrospun PIM-EA-TB fibres greatly depends on the chloroform/n-PL proportion.

With pure chloroform, dumbbell-like fibres without beads were obtained (Figure 2a). With the addition of n-PL into the binary solvent system, the electrospinning yielded fibres with a cylindrical shape. For the sample 9:1, dumbbell-shaped fibres were still obtained (Figure 2b), then increasing the proportion of n-PL in the binary solvent system, the cross-section of the fibres became circular (Figure 2c,d). The formation of dumbbell-like fibres is commonly observed when highly volatile solvent systems are used during the electrospinning [5,32] due to the collapse of the fibre skin during the rapid vaporisation of solvent. When n-PL is added, the mixture becomes less volatile and less prone to collapse. However, as soon the content of n-PL exceeded chloroform content, the presence of n-PL induced fused fibre formation due to insufficient vaporisation of the solvent molecules from the jet. When pure n-PL is used, the electrospinning was not possible, only spraying occurred (Figure 2f). It is also important to note that the dumbbell shape is not affected by the process parameters, as this shape has been observed at several gap, flow and voltage values. Only the presence of n-PL affected it.

Another interesting feature was the porous surface. The SEM images of electrospun PIM-EA-TB fibres with higher magnifications are given in Figure 3.

It was observed that when pure chloroform was used as solvent for the electrospinning, porosity appeared on the surface of the fibres, with a 150 nm pore diameter (Figure 3). On the contrary, when n-PL was added, the porous surface disappeared. The pore formation is due to a local phase separation on the surface [32]. The fast evaporation of solvent leads to local phase separation, and the solvent-rich regions transform into pores during the electrospinning process. With the addition of n-PL, the boiling point of the solvent system is higher than chloroform alone. This induces a slowdown of the solvent evaporation, and thus eliminates pore formation.

Fibre diameter

The presence of n-PL induced a reduction in the fibre diameter as well. As reported in Table 2, the fibre diameter decreased from 7.9 μm to 4.7 μm for the sample 10:0 and the sample 5:5, respectively.

This decrease is related to the electrical conductivity variation. As showed in Figure 4, the diameter of the fibre decreased with an increase in the conductivity of the solution. The increase in conductivity induces greater stretching of the electrospinning jet and favours a reduction in fibre diameter [33].

#### 3.1.2. Effect of the Electrospinning Process Conditions

The influence of the process conditions on the fibre diameter was further studied for the solutions yielding to bead-free structures only, i.e., systems with 10:0, 7:3, and 5:5 as CH_3_Cl/n-PL ratio.

Effect of flow rate

Different feed flow rates, from 10 to 100 μL min^−1^, were studied for each of the solutions. Figure 5a shows the impact of the flow rate on the fibre diameter. The mean diameters of the microfibres increased from 3.2 ± 1, 1.8 ± 0.5 and 1.7 ± 0.5 μm to 7.9 ± 1, 5.3 ± 0.5 and 4.7 ± 0.5 μm to as a result of the increase in the flow rate from 10 to 100 μL min^−1^ for 10:0 sample, 7:3 sample and 5:5 sample, respectively. A higher flow rate results in thicker fibres due to the higher mass flow through the tip.

Effect of gap

Three different distances between tip and collector were used to study their impact on the fibre diameter (Figure 5b). The mean diameters of the microfibres decreased from 4.1 ± 1, 3.4 ± 0.5 and 3.7 ± 1 μm to 3.1 ± 1, 2.8 ± 0.5 and 2.4 ± 0.5 μm to as a result of the increase in the gap from 10 to 30 cm for the 10:0 sample, 7:3 sample and 5:5 sample, respectively. By increasing the gap, thinner fibres were formed for the three systems since a higher distance between the tip and the collector provides a larger stretching distance.

Effect of voltage

Finally, the effect of the applied voltage was studied for each solutions and shown in Figure 5c. A minimum voltage was required in order to form fibres. Below 16 kV, fewer fibres were obtained. By increasing the voltage, the electrospinning yield was enhanced with the formation of microfibres. The mean diameters of the microfibres decreased from 4.9 ± 1, 3.9 ± 1 and 3.6 ± 1 μm to 2.7 ± 1, 3 ± 0.5 and 3.3 ± 0.5 μm to as a result of the increase in the applied voltage from 16 to 25 kV for 10:0 sample, 7:3 sample and 5:5 sample, respectively. Actually, increasing voltage induced a larger stretching of the solution resulting in the formation of thinner fibres [12,26,34,35].

The influence of applied voltage is decreasing with the addition of n-PL. For the system 10:0, the fibre diameter is divided by 45%, whereas for the 5:5 system the fibre diameter is only reduced by 11%. The diameter reduction is determined by the strength of the electric field [11]. As n-PL is more conductive than chloroform (Table 1), the strength of the electric field is lower with the n-PL solutions inducing a smaller reduction in fibre diameter.

### 3.2. Characterisation

#### 3.2.1. Thermal Analysis

Figure 6 shows the Thermogravimetric Analysis (TGA) graphs of PIM-EA-TB fibres obtained from the solution (10/0), (9/1), (7/3) and (5/5).

The TGA curves show a different behaviour for the fibres obtained with pure chloroform and the ones with the mixture of n-PL/chloroform. The fibres obtained from (10/0) solution present a slight first degradation at Td_1_ = 40 °C due to the presence of residual chloroform. Then, a second degradation temperature, Td_2_, is observed around 325 °C corresponding to the Td of PIM-EA-TB [29].

As soon n-PL is added in the solution, a larger degradation is observed at Td_1_, which is shifted to higher value, 120 °C, due to the higher boiling point of n-PL compared to chloroform (Table 1). The second degradation is also observed at the same temperature of 325 °C.

#### 3.2.2. Pore Size

The fibrous mats were also characterised in terms of porosity (Figure 7a).

The fibrous mats had a pore diameter varying from 11 μm to 51 μm depending on the binary solvent mixture. The fibres obtained from pure chloroform solution gave the larger pore diameter. The solutions with n-PL, 7:3 and 5:5 had similar smaller pore diameters, from 11 μm to 28 μm.

Regardless of the binary system used as solvent, the electrospun fibres exhibit a similar correlation between fibre diameter and pore diameter, with an increase in fibre diameter inducing an increase in pore diameter (Figure 7a).

#### 3.2.3. Air Resistance

As indicated above (Section 2.3.2), it is preferred to use the airflow resistance than the air permeability, meaning that a high value of R_Air_ represents a low permeable material, whereas a low value of R_Air_ is representative of a highly permeable material. The PIM-EA-TB fibrous mats exhibited R_Air_ from 25 × 10^7^ m^−1^ to 5 × 10^7^ m^−1^, which are similar to standard values of R_Air_ for commercial facemasks [12]. As expected, the air flow resistance decreased as the pore diameter increased (Figure 7b). The sample from pure chloroform was the sample with the lowest R_Air_ due to the fact it had the larger pore diameter. Similar to the pore size variation, the airflow resistance presents the same trend for the three binary mixtures regardless the solvent used during electrospinning.

For a same pore size diameter, the fibres obtained from the chloroform solution present a higher R_Air_ than the ones from binary solutions. The dumbbell shape might add a supplementary resistance to the air transport though the fibrous mat.

#### 3.2.4. Water Resistance

The water contact angle of the electrospun fibres was characterised (Appendix A). The fibrous mats present a high hydrophobic character, with a water contact angle around 126°. The hydrophobic character is related to the high surface roughness of the fibrous surface.

The water resistance property was also investigated by hydrostatic pressure tests. The hydrostatic pressure of liquid water corresponds to the pressure required to penetrate the sample and form water droplets on its opposite surface. A larger value means higher resistance to liquid water penetration. As suggested by Figure 7c, the hydrostatic pressure followed the same trend as R_Air_, it decreased with increasing pore size diameter. Actually, a large pore diameter induces a lower resistance to water penetration. The fibres from chloroform solutions appear to be more resistant to water than the ones from the solutions with n-PL.

Waterproofness, i.e., water resistance and sweat water breathability was also tested for the electrospun fibres. A fibrous mat was put over a beaker filled with boiling water with silica particles and a droplet of water on the surface (Figure 8).

The silica particles were used to demonstrate the water vapour transmission and so the ability of the sample to release the moisture vapour, whereas the water droplets were used to demonstrate waterproofness. For all the samples, the water droplets were still present on the surface after more than 1 h, indicating the water resistance of the fibrous mat. After 30 min, water vapour appeared on the top surface of the sample (Figure 7b), and the silica particles started to change colour meaning that water vapour was transported through the mat confirming the breathability of the mat. This simple experiment coupled with R_Air_ data demonstrates the potential of PIM-EA-TB fibres to be used as facemasks.

## 4. Conclusions

This paper reports for the first time the fabrication of PIM-EA-TB fibres with controlled morphology using a chloroform/n-Propyl lactate solvent solution with different volume ratios. Different surface morphologies (porous surface), shapes (cross-sectional dumbbell shape) and diameter were obtained according to the solvent system due to their physical properties (conductivity, boiling point). Firstly, the use of a volatile solvent, such as chloroform, induced the formation of a dumbbell shape and a surface porosity. With the addition of n-PL, which presents a higher boiling point, circular fibres were obtained. Secondly, formation of thick fibre was observed with low conductive solvent systems, with diameter up to 7.9 μm. With the addition of a more conductive solvent, a decrease in the diameter fibres was noticed, with diameter down to 4.7 μm. In terms of performances, the fibrous mats obtained with pure chloroform presented smaller pore size, higher air and water resistance than with n-PL due to the dumbbell shape.

## Figures and Tables

**Figure 1 membranes-11-00422-f001:**
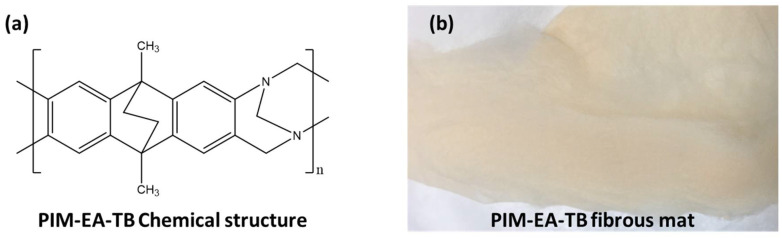
(**a**) Chemical structure of Polymer of Intrinsic Microporosity derived from ethanoanthracene and Tröger’s base (PIM-EA-TB). (**b**) Electrospun fibrous mat of PIM-EA-TB.

**Figure 2 membranes-11-00422-f002:**
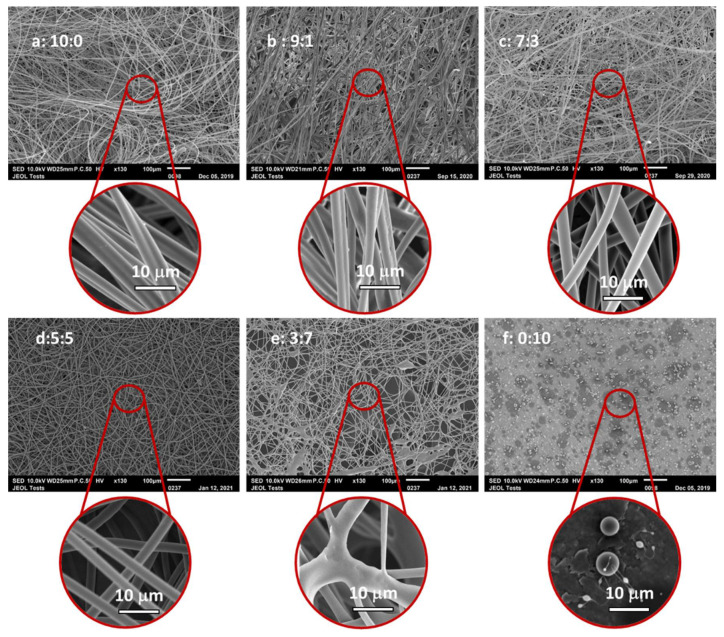
Scanning Electron Microscopy (SEM) micrographs of PIM-EA-TB fibres from solutions varying the proportion of chloroform/n-PL. (**a**) = 10:0; (**b**): 9:1; (**c**): 7:3; (**d**): 5:5; (**e**): 3:7, (**f**):0:10. (Conditions: (**a**,**b**) Flow = 100 μL/min, Gap = 30 cm, Voltage = 25 kV; (**c**–**f**) Flow = 100 μL/min, Gap = 20 cm, Voltage = 16 kV).

**Figure 3 membranes-11-00422-f003:**
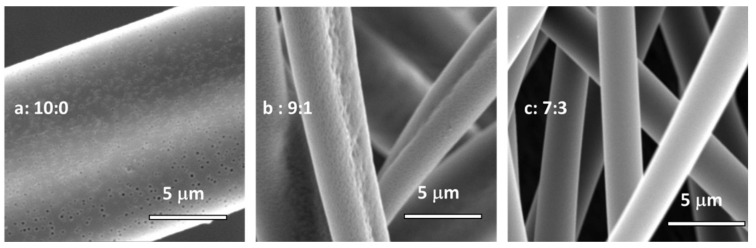
SEM micrographs at high magnification of PIM-EA-TB from 10:0 (**a**), 9:1 (**b**) and 7:3 (**c**) chloroform/n-PL solutions. (Conditions: (**a,b**) Flow = 100 μL/min, Gap = 30 cm, Voltage = 25 kV; (**c**) Flow = 100 μL/min, Gap = 20 cm, Voltage = 16 kV).

**Figure 4 membranes-11-00422-f004:**
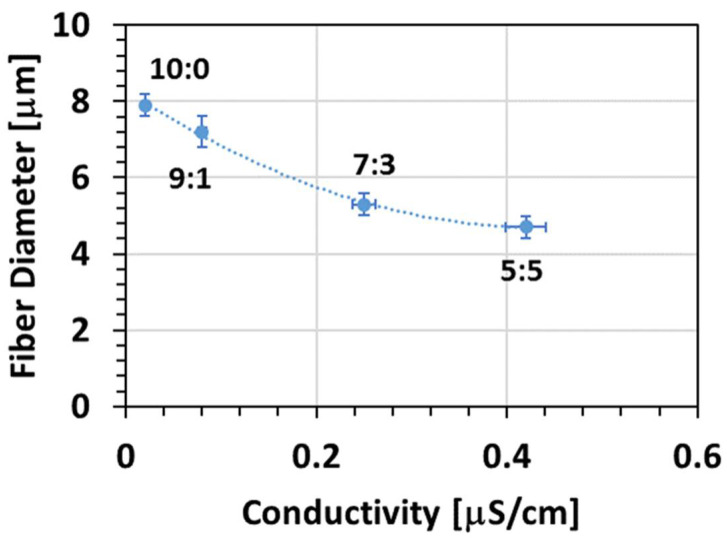
Variation in the fibre diameter with the increase in the conductivity from solutions varying the proportion of chloroform/n-PL. (Conditions: (10:0, 9:1) Flow = 100 μL/min, Gap = 30 cm, Voltage = 25 kV; (7:3, 5:5) Flow = 100 μL/min, Gap = 20 cm, Voltage = 16 kV).

**Figure 5 membranes-11-00422-f005:**
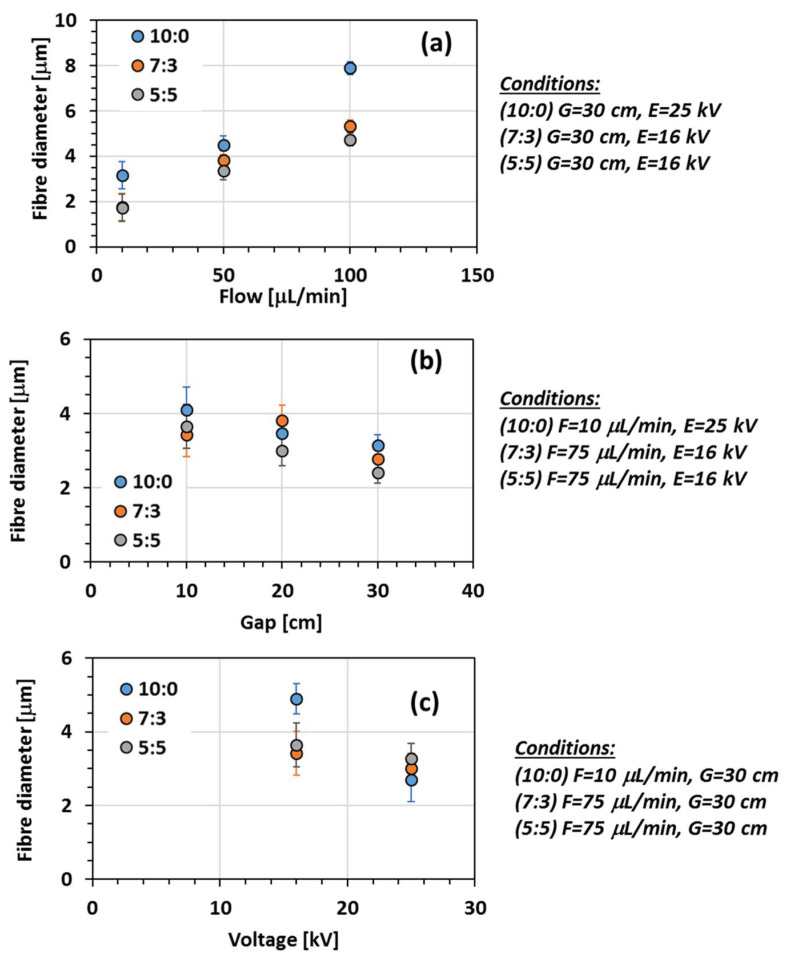
Impact of flow rate (**a**), gap (**b**) and voltage (**c**) on fibre diameter of PIM-EA-TB from solutions varying the proportion of chloroform/n-PL.

**Figure 6 membranes-11-00422-f006:**
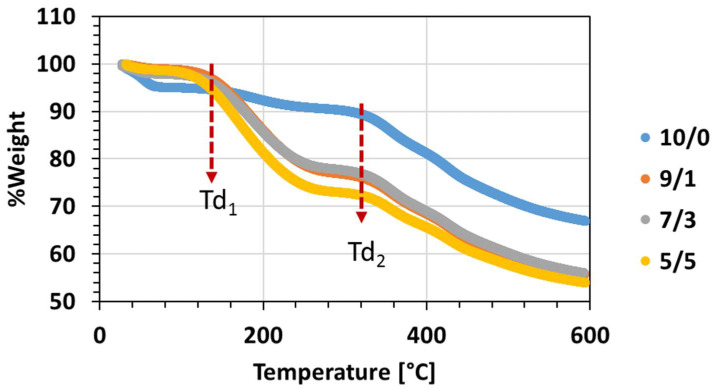
Thermogravimetric Analysis (TGA) of PIM-EA-TB electrospun fibres from solutions (10/0), (9/1), (7/3) and (5/5).

**Figure 7 membranes-11-00422-f007:**
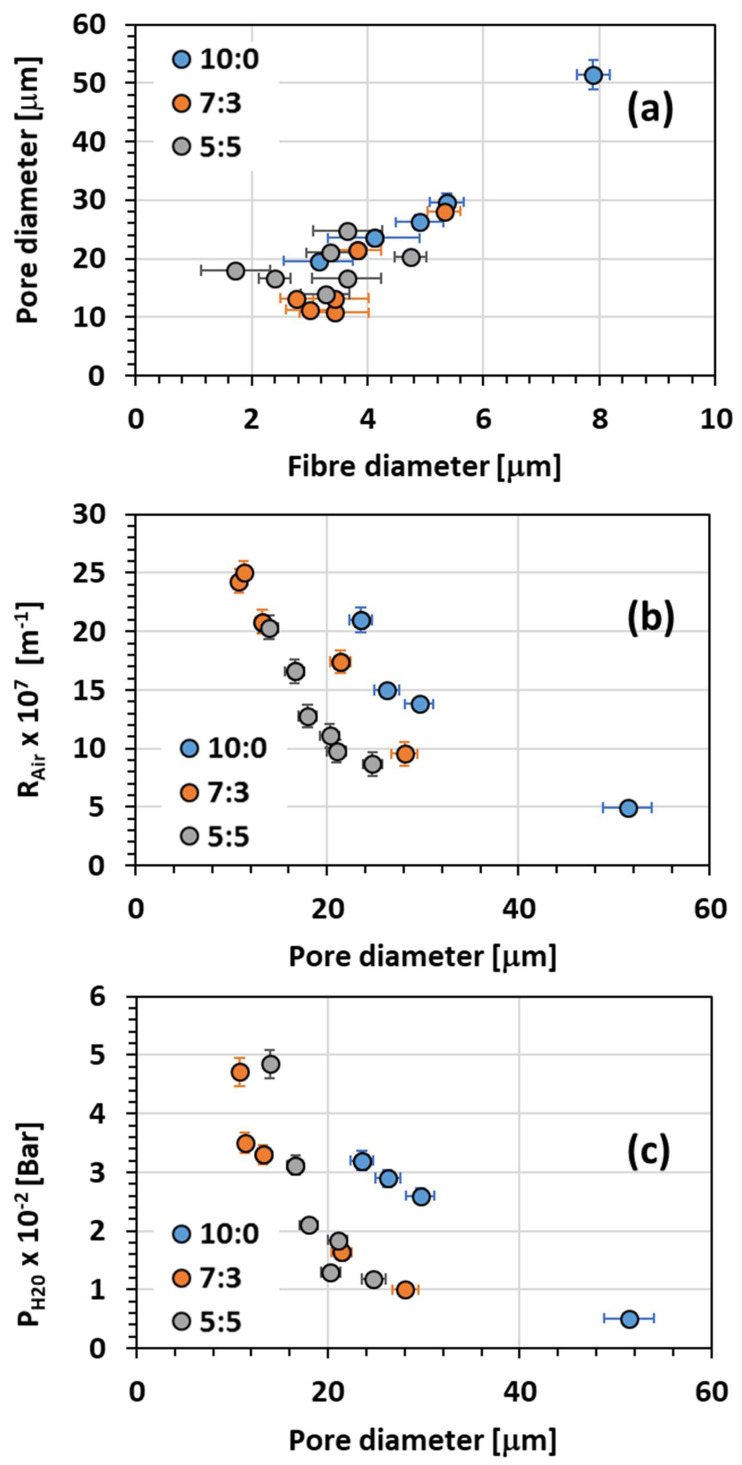
(**a**) Variation of pore diameter with fibre diameter, (**b**) variation of air flow resistance (R_Air_) with pore diameter and (**c**) variation of hydrostatic pressure (P_H2O_) with pore diameter for PIM-EA-TB fibres from solutions varying the proportion of chloroform/n-PL.

**Figure 8 membranes-11-00422-f008:**
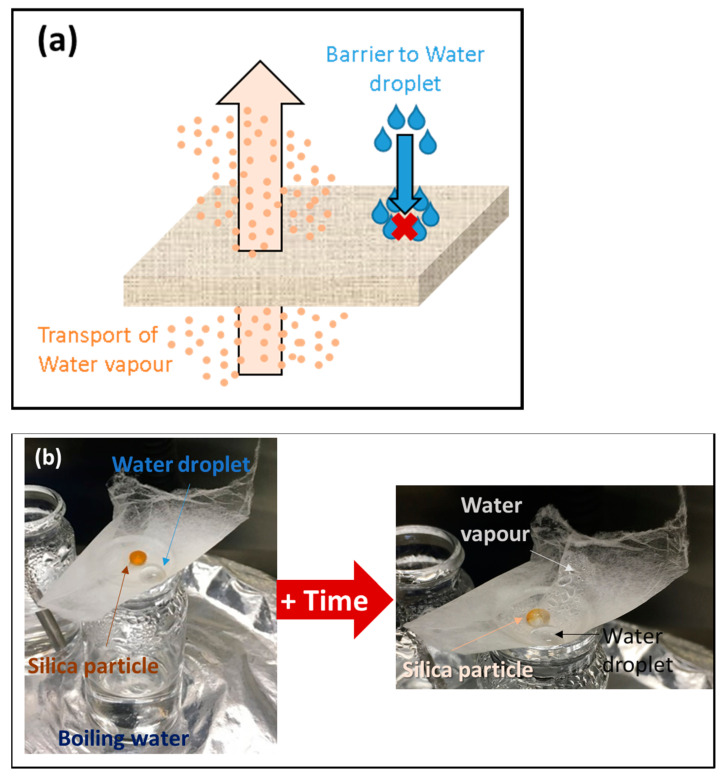
(**a**) Schematic mechanism and (**b**) experimental test for waterproofness and breathability performance (Sample 5:5).

**Table 1 membranes-11-00422-t001:** Physical properties of the solvents (a) obtained from MSDS and (b) measured.

Solvent	Chloroform	n-Propyl Lactate
Boiling point (°C)	61 ^(a)^	170 ^(a)^
Viscosity at 20 °C (10^−3^ Pa·s)	0.56 ^(a)^	3.3 ^(a)^
Electrical conductivity at 20 °C (S·cm^−1^)	1 × 10^−11 (a)^	2 × 10^−6 (b)^

**Table 2 membranes-11-00422-t002:** Fibre diameter of PIM-EA-TB fibres from (10:0), (9:1), (7:3) and (5:5) chloroform/n-PL solutions.

Ratio Chloroform/n-PL	10:0	9:1	7:3	5:5
Fibre diameter (μm) ± 0.3 μm	7.9	7.2	5.3	4.7

## Data Availability

The data presented in this study are available on request from the corresponding author. The data are not publicly available due to privacy.

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
