# Peer review of "Control Over the Morphology of Electrospun Microfibrous Mats of a Polymer of Intrinsic Microporosity"

_membranes, 2021, doi:10.3390/membranes11060422_

Round 1

Reviewer 1 Report

In my opinion, the manuscript can be published after major revision for the following reasons.

In figure 4 were reported the impact of the process parameter of electrospinning on fiber diameters varying the proportions of solvents. But from the methodological point of view (see Fig.4a) in order to evaluate the influence of each parameter on fiber diameter should be varied one parameter for time, instead, the authors compare the different systems varying the flow and also the applied voltage, also in the Fig. 4b the conditions of the electrospinning for different systems is not the same in terms of the flow rate and applied voltage. This is probably due to the different viscosity of raw solutions, for this, the measurement of viscosity of solutions to electrospun should be added. 

The most important features of  PIMs, ie the surface area and dimensions of the micropores, are not investigated.

Moreover, in order to evaluate the potential applications, the thermal and mechanical stability of electrospun materials should be investigated.

Minor revision:

When the acronyms appear for the first time, should be explained (i.e. abstract row 11).

The samples compared in Figure 2 and Figure 3 in which conditions, in terms of applied voltage and GAP, were obtained?

In figure 4, was reported 1:0 instead of 10:0.

Reviewer 2 Report

Review of Membranes « control cover the morphology… microporosity”

The manuscript, entitled “Control over the morphology of electrospun microfibrous mats of a polymer of intrinsic microporosity”, describes the effects of the solvent choice during the electrospinning process on the fibers morphology and physical properties of a polymer of intrinsic microporosity. The paper is relatively well written and organized, and underlines the influence of the process conditions on the morphology and porosity of the electrospun mats.

The study is very interesting, nevertheless, it would deserve to be completed in particular with the analysis of the properties of the various tested solutions to justify the conclusions made during the analysis.

Comments

  • The introduction part should be revised and organized in 4 paragraphs (generalities, state of the art, lacks in the state of the art, and content of the present work).The presence of figure in the introduction should be avoided. It seems that Figure 1b is not cited in the text.
    The introduction should also underline the interest of the kind of study, and the desired properties of the mats for further application
  • Page 2 – Table 1 shows the physical properties of the selected solvents, but the physical properties of the various working solutions are missing, it should be added or measured to analyze the effect of these on the electrospinning process.
  • Part 3.1.1. The electrospinning parameters should be added in the discussion, or at least in the legend caption of the figure 2. The formation of the dumbbell shape is it related only to the solution properties or also to tip to collector distance, flow rate, or voltage? Why the authors have chosen 20% for the concentration? The obtained morphology should be also discussed in regard to the physical properties of the solutions (viscosity, surface tension, conductivity, polymer entanglement macromolecular chains).
  • Page 5 – Table 2 – standard deviation should be added
  • P6 – Figure 4 - The parameters are not the same for the tested samples (Figure 4-a & b, E is different, Figure 4-c, it is the flow rate that differs), this choice of conditions must be explicitly explained, and justified in terms of limits.

Sometimes it is 10:0, other times 1:0 – please correct

  • P7 – line 195 “as nPL is more conductive than chloroform…”, to justify the effect of the conductivity of the raw solvent, the conductivity of each solution should be measured
  • Part 3.2.3 – the surface properties of the mat may be also determined by contact angle measurements
  • Conclusion should be more detailed.

Round 2

Reviewer 2 Report

In this revised version, most of the comment have been taken into account to improve the paper quality